# Mild Pretreatments to Increase Fructose Consumption in *Saccharomyces cerevisiae* Wine Yeast Strains

**DOI:** 10.3390/foods10051129

**Published:** 2021-05-19

**Authors:** Hatice Aybuke Karaoglan, Filiz Ozcelik, Alida Musatti, Manuela Rollini

**Affiliations:** 1Department of Food Engineering, Sivas Cumhuriyet University, Sivas 58140, Turkey; akaraoglan@cumhuriyet.edu.tr; 2Department of Food Engineering, Ankara University, Ankara 06830, Turkey; filiz.ozcelik@ankara.edu.tr; 3Department of Food, Environmental and Nutritional Sciences (DeFENS), Università degli Studi di Milano, 20133 Milano, Italy; alida.musatti@unimi.it

**Keywords:** glucose consumption, fructose consumption, *Saccharomyces cerevisiae*, sigmoidal model, pretreatment, developed Gompertz model

## Abstract

The present research investigates the effect of different pretreatments on glucose and fructose consumption and ethanol production by four *Saccharomyces cerevisiae* wine strains, three isolated and identified from different wine regions in Turkey and one reference strain. A mild stress temperature (45 °C, 1 h) and the presence of ethanol (14% *v*/*v*) were selected as pretreatments applied to cell cultures prior to the fermentation step in synthetic must. The goodness fit of the mathematical models was estimated: linear, exponential decay function and sigmoidal model were evaluated with the model parameters R^2^ (regression coefficient), RMSE (root mean square error), MBE (mean bias error) and χ^2^ (reduced Chi-square). Sigmoidal function was determined as the most suitable model with the highest R^2^ and lower RMSE values. Temperature pretreatment allowed for an increase in fructose consumption rate by two strains, evidenced by a t90 value 10% lower than the control. One of the indigenous strains showed particular promise for mild temperature treatment (45 °C, 1 h) prior to the fermentation step to reduce residual glucose and fructose in wine. The described procedure may be effective for indigenous yeasts in preventing undesirable sweetness in wines.

## 1. Introduction

Microorganisms have evolved a number of mechanisms in response to external environmental changes or stresses, allowing them to successfully tolerate modifications and to rapidly adapt in order to maintain cell integrity with an efficient metabolic activity. During alcoholic fermentation, yeasts are directly exposed to a changing environment so they have to maintain close-to-optimal conditions within the organism as a whole, evidenced by accurate measures of kinetic parameters and sugar consumption capacity [1]. While unstressed cells exhibit moderate resistance to a range of environmental stresses, mildly stressed cells significantly increase their ability to withstand future insults [2,3]. This adaptation may be advantageous for yeasts metabolism, depending on the severity of the change in environmental conditions. Piper et al. [4] showed that exponentially grown cultures of *Saccharomyces cerevisiae* lead to cell death following a first order process when suddenly exposed to 50 °C; nevertheless, exposure to milder conditions (37 °C) leads to a transient growth arrest, demonstration of an adaptive response followed by a growth resumption.

In a wine must, sugar concentration is in the range 18–25% *w*/*w*, even if standard deviation can be up to 2% *w*/*w*. Sugars are the most abundant compounds, in the form of fructose and glucose, in near equimolar amounts. Grapes also contain organic acids, largely retained in must, and the resulting low pH excludes the growth of many spoilage and pathogenic microorganisms. Tartaric acid is generally the major acid, followed by citric and malic acids. The amino acids can vary depending on grape variety; sometimes, musts may be supplemented with nitrogen, typically diammonium phosphate [5]. Yeasts, under anaerobic conditions, consume nitrogen in the forms of ammonium cations, amino acids and small peptides: these are called YAN, yeast assimilable nitrogen. YAN composition in must depends on different factors, for example the cultivar, the maturity of grapes, as well as the cultural practices in the vineyard [6].

During fermentation, generally, *S. cerevisiae* has a slight preference for glucose, resulting in a difference of their consumption [1,7,8]. This preference gives rise to a discrepancy between the amount of glucose and fructose consumed during fermentation, with fructose being the predominant residual sugar [9]. The dominant effect of glucose on yeast carbon metabolism is coordinated by several signaling and metabolic interactions that mainly regulate transcriptional activity but are also effective at post-transcriptional and post-translational levels [10]. This effect is reported to be one of the causes of arrested or so-called stuck fermentation [11]. Moreover, since fructose is approximately twice as sweet as glucose, excess fructose can cause undesirable sweetness in wines [12]. So far, it is interesting to investigate the kinetic aspects related to glucose and fructose consumption rates, in order to optimize the fermentation process and produce wines with a low residual fructose level. Searching for yeast strains with higher fructose preference indeed is an important goal for winemakers [13].

The development of improved kinetic models of wine fermentation may provide better process prediction from the beginning to the end of fermentation, the final goal being the decrease in energy costs [14]. Most of the models, consisting of a series of mathematical equations describing the events that occur during wine fermentation, are biochemically knowledge-based models. Boulton [15] developed the first model to examine the effect of factors, such as sugar concentration and temperature, during fermentation. A similar mechanical model was developed for the identification of microbial activities in grape must fermentation, such as product synthesis other than respiration and ethanol production [16]. However, none of these models was able to acceptably predict fermentation problems, including process temperature and variations of the initial conditions [17]. Models defining important biological parameters can be obtained; however, their structures can be strongly nonlinear, complex and difficult to verify, also taking into account the difficulties in correctly defining the parameters. 

In this frame, the use of sigmoidal shaped growth models in the field of descriptive microbiology to predict the fermentation process in the development of nonlinear modeling techniques is increasing [18]. Within the nonlinear models, the Gompertz equation is an example of exponential equations. Based on the assumption that the probability of survival of microorganisms increases exponentially over time, this experimental function defines growth in limited nutrient-containing environments. Tronchoni et al. [19], in research on the modelling of substrate consumption during fermentation of grape juice, stated that the Gompertz model was particularly weak in defining the final parts of the fermentation; authors highlighted that numerous yeasts showed a sigmoid-decay response for glucose and fructose consumption at 12 °C.

In this study, the differences in the fermentation performance of *S. cerevisiae* wine strains subjected to pretreatments before the fermentation were determined in a synthetic medium, possessing a very similar composition to grape must. Pretreatments may have the final aim of modifying sugar consumption or ethanol production, thus limiting undesirable wine tastes due to residual fructose. In order to identify and predict fermentation process in *S. cerevisiae* wine strains, the compatibility of sugar consumption data was compared applying different mathematical models. For this purpose, the suitability of linear, exponential and sigmoidal models was determined; the t50 and t90 values, defined as the time required to use 50 and 90% of the glucose and fructose in the environment, were also calculated.

## 2. Materials and Methods

### 2.1. Yeast Strains and Culture Conditions

Three indigenous *S. cerevisiae* were used in the research, as follows: *S. cerevisiae* Kalecik II (coded 2), *S. cerevisiae* Narince 3 (coded 7) and *S. cerevisiae* 1A (coded 47). One commercial wine strain was used as reference: Fermicru AR2 *S. cerevisiae* No. LO122 (DSM Food Specialties, Holland, coded S122). Indigenous strains were isolated from Turkish wines and obtained from Ankara University culture collection, isolated and identified as reported elsewhere [20].

Strains were maintained in YPD broth (Sigma Aldrich, Steinheim, Germany), grown at 30 °C and subsequently stored at 4 °C until use. 

### 2.2. Pretreatments Application

Approximately 18 h pre-grown strains in YPD broth were taken (1 mL) and inoculated into 10 mL YPD broth, monitoring the growth curve with a spectrophotometer (Optima, SP 300, Tokyo, Japan) at OD 600 nm. When the culture reached the mid log phase (approximately 4 h), cells were exposed to the two selected pretreatments: temperature (45 °C, 1 h) and presence of ethanol (14% *v*/*v*), comparatively. Temperature pretreatment was applied employing a water bath, as follows: culture was placed in a water bath set at 60 °C in shaking conditions in order to increase the temperature as early as possible, controlling culture temperature using an un-inoculated tube containing 10 mL YPD broth; when the broth reached 45 °C (approximately 1–2 min), cultures were taken into another water bath set at 45 °C and incubated for 1 h with occasional stirring. After pretreatment, 1 mL of this culture was immediately inoculated into the fermentation must (10 mL).

Ethanol pretreatment was applied as follows: ethanol was added to the culture to reach a final concentration of 14% (*v*/*v*), shaking vigorously. Immediately after, 1 mL of culture was taken and inoculated into the fermentation must. Untreated culture samples (control) were also grown in the same manner, but they were kept at +4 °C during the pretreatments.

### 2.3. Fermentation Trials

Fermentation trials were carried out in duplicate in synthetic must, having the following composition (g/L): glucose 115, fructose 115, yeast nitrogen base (with ammonium sulfate without amino acids, Difco, Carrickmore, UK) 6.7, citric acid 0.2, malic acid 3.0, tartaric acid 5.0. The medium was adjusted to pH 3.5 with 0.1 N HCl and sterilized with 0.22 µm filter (Sartorius, Germany).

Fermentations were carried in 100 mL glass flasks containing 10 mL of synthetic must; flasks were equipped with an airlock system containing 3 mL 10% H_2_SO_4,_ in order to ensure the anaerobic environment. For pretreated samples, 10 mL of synthetic must were inoculated with 1 mL of pretreated YPD culture, ensuring that yeast cell concentration was almost the same for all trials at the beginning of fermentation. Fermentation trials were all performed in duplicate at 30 °C in static condition. 

### 2.4. Analytical Determinations

At appropriate intervals and for each fermentation trial, cell concentration as well as glucose, ethanol and fructose concentrations were determined.

Cell concentration was calculated by taking aliquots from the fermentation samples and, after decimal dilutions, inoculating them on YPD medium by the surface spreading method; plates were incubated for 2 days at 30 °C, and then the numbers of colonies were counted (log CFU/mL). 

Sugars and ethanol quantification was performed as follows: 1 mL fermentation sample was taken and centrifuged in Eppendorf tubes for 10 min at 14,500× *g* (Microspin 12, Biosan, SIA, Riga, Latvia); supernatants were separated and kept at −18 °C for sugars and ethanol analyses. Concentrations were determined using an HLPC system (L 7000, Merck Hitachi, Darmstadt, Germany) equipped with serially connected UV and RI detectors using a SH1821 (300-8 mm) (Shodex, Munich, Germany) column, maintained at 50 °C and eluted with 5 mM H_2_SO_4_ at 0.5 mL/min [21]. 

### 2.5. Mathematical Modelling of Glucose and Fructose Consumption

The suitability of different mathematical models to define sugar consumption by yeasts was checked and compared. For this purpose, the level (%) of residual glucose and fructose was calculated at each sampling time. The models and equations used in the study are here reported:-Linear equation [7]:
(1)$$Y=S_{0}-k\times t$$
where *Y* represents the residual glucose and fructose (%) in the medium, *S*_0_ the intercept value, *k* the kinetic constant of linear equation (concentration/h) and *t* the fermentation time (h).
-Exponential function model [13]:
(2)$$Y=D+S_{0}\times e^{-K\times t}$$ where *Y* represents the residual glucose and fructose (%) in the medium, *D* the specific value when *t*→∞, *S*_0_ the intercept value, *k* the kinetic constant of exponential function model (h^−1^) and *t* the fermentation time (h).
-Sigmoidal or modified Gompertz function model [22]:
(3)$$Y=A+C\times e^{-e^{k\times \left(t-M\right)}}$$ where *Y* represents the residual glucose and fructose (%) in the medium, *A* the lower asymptote when *t*→∞, *k* the kinetic constant of the sigmoidal model (h^−1^), *M* the time when the inflection point is obtained (h) and *t* the fermentation time (h).

For each model, the regression coefficient (R^2^) gives the correlation between observed (average of two data for each treatment) and predicted data; the root mean square error (RMSE) indicates the actual deviation between observed and predicted data; the mean bias error (MBE) refers to model overestimation or underestimation; the reduced mean square of the deviation or the reduced chi-square (χ^2^) belonging to the 3 different mathematical models were also compared. 

The equations of RMSE, MSE and χ^2^ are given below:(4)$$RMSE={\left[\frac{1\;}{N}\overset{N}{\underset{i=1}{\sum }}{\left(C_{pre.}-C_{obs.}\right)}^{2}\right]}^{1/2}$$
(5)$$MBE=\frac{1\;}{N}\overset{N}{\underset{i=1}{\sum }}{\left(C_{pre.}-C_{obs.}\right)}^{2}$$
(6)$$\chi^{2}=\frac{\sum_{i=1}^{N}{\left(C_{pre.}-C_{obs.}\right)}^{2}}{N-z}$$
where *N* represents the number of observations, *C* the residual glucose and fructose (%) in the medium (predicted and observed) and *z* the number of constants used in the model.

### 2.6. Statistical Analysis

The effect of factors on dependent variables was determined by analysis of variance using SPSS Statistics for Windows program (v. 16.0, Armonk, NY, USA, IBM Corp.). The statistically significant differences by treatments were determined with Tukey test using the same program. Correlation analyses were carried out using the Statistica**^®^** package program (1995, StatSoft, Tulsa, OK, USA).

## 3. Results

Pretreatments, considered as mild stress applications, were applied to three indigenous and one reference *S. cerevisiae* strains (coded 2, 7, 47 and S122, respectively), monitoring glucose and fructose consumption as well as ethanol production in the fermentation medium up to 48 h.

All tested strains reached the mid-log phase in 4 h approximately. In fermentation trials, yeast concentration did not change in the first 4 h (lag phase), then it rapidly increased between 4 and 19 h (log phase) to reach the stationary phase before 19 h for strains 2 and S122 independently from the treatment applied; strains 7 and 47 instead entered the stationary phase at 19 or 24 h depending on the treatment (Figure 1). The initial pitching rate was around 6 log CFU/mL, typical of a grape must fermentation, and in all trials, the maximum cell concentration was always found around 8 log CFU/mL.

Ethanol was detected from 19 h onward, for all strains and treatments. The effects of different pretreatments on glucose, fructose and ethanol concentrations were evaluated in three different periods, expressed as 0–19, 0–24 and 0–48 h: 19 h was the first time in which ethanol appeared, 24 h the middle of the fermentation process and 48 h the moment in which the fermentation stopped. These three periods were formed by taking the absolute differences between glucose, fructose and ethanol levels at the beginning of fermentation and the considered time, as expressed in Table 1.

In the period defined as 0–48 h, strains consumed glucose in the range 90.05–102.18 g/L (control), 80.58–103.18 g/L (temperature treated) and 77.53–127.35 g/L (ethanol treated), respectively. Fructose was instead consumed at lower amounts, in the range 75.38–90.45 g/L (control), 60.18–98.15 g/L (temperature treated) and 66.73–87.93 g/L (ethanol treated), confirming the literature findings about *S. cerevisiae*’s preference for glucose consumption [9,12]. Ethanol was produced in the range 66.93–92.43 g/L (control), 58.70–75.89 g/L (temperature treated) and 70.45–87.30 g/L (ethanol treated), respectively. Note that the highest fructose consumption (98 g/L) was evidenced for strain 7 applying the temperature pretreatment.

The ANOVA test was used to determine the effect of the type of strain, different time periods and type of pretreatment on glucose and fructose consumption and ethanol production. Results showed that the strain difference was only effective on glucose consumption; as expected, different time periods were effective on all dependent variables, type of pretreatments was effective on all dependent variables, and the interaction of type of pretreatments x strain was effective on glucose and fructose change (*p* < 0.05) (Table 2). In order to better understand the interaction between the type of treatment and the type of strain on glucose and fructose consumption, the difference was examined with the Tukey test (Table 3). Among the strains subjected to temperature pretreatment, a statistically significant difference was determined between 7 and S122 strains on glucose consumption, the first being the most glucose-consuming strain, while the S122 was the least. Results also highlighted that the highest fructose consumption (residual < 50 g/L) was evident for the indigenous temperature-pretreated strain 7 and for the reference ethanol-pretreated strain S122.

In order to understand the relationship between the amount of residual glucose and fructose in the environment during the fermentation period, the Pearson correlation was used. The amount of residual glucose and fructose in the fermentation must for all yeasts was found to be inversely correlated with the duration of the fermentation trial: each unit increase in time caused a 2% glucose (Pearson correlation: r −0.95, *p* < 0.001) and 1.7% fructose (Pearson correlation: r −0.93, *p* < 0.001) decrease. 

Glucose and fructose consumption in all fermentation conditions was parametrized applying mathematical models and goodness of fits compared with the values R^2^, RMSE, MBE and χ^2^ (Table 4). As regards glucose consumption, R^2^ values for linear and exponential function models were found between 0.806–0.986 and 0.763–0.994, respectively; R^2^ values belonging to the sigmoidal function model were instead higher, in the range 0.991–0.999. RMSE values, indicating the difference between the estimated and the observed data in this last modelling, were found to be at most 4.524, while MBE and χ^2^ were always lower, 1.712 and 40.94, for glucose consumption with a sigmoidal model. 

As regards fructose consumption, R^2^ and RMSE values were found in the range 0.795–0.976 and 4.00–20.18 for linear model, respectively. R^2^ of exponential function could not be determined for all the tested conditions, and also, RMSE values were very high. Applying the sigmoidal model, R^2^ values obtained after the treatment were high and at least 0.964, while the highest RMSE was 5.709. 

When the goodness of data fit was evaluated, the sigmoidal model was the most suitable to explain the use of glucose and fructose consumption. Figure 2 clearly shows that data estimated by the sigmoidal model for glucose and fructose consumption fit pretty well with the observed data for all treatments and strains. 

Table 5 shows the entire set of equations created using the sigmoidal model that can be applied to evaluate the percentage of residual glucose and fructose employing untreated and pretreated yeasts samples; also, t50 and t90 values, i.e., the time needed to consume 50 and 90% of the initial sugar content, are presented. 

According to these data, as regards glucose consumption, t50 and t90 were found in the range 21.67–24.60 and 36.39–49.22 h, respectively, for control samples, 25.64–34.60 and 33.39–43.71 h for temperature-pretreated samples and 18.38–22.60 and 34.52–49.13 h for ethanol-pretreated samples, respectively. The effect of strain difference on t50 and t90 values was found to be significant, while the effect of pretreatment was significant only for t50 (*p* < 0.05). Ordering the strains from the most performing (glucose consumption) to the least efficient, the rank for t50 values is as follows: 47, S122, 7 and 2; for t90 values, the difference between the most performing strains 47 and S122 and the least 7 and 2 were not significant, but the difference between these two groups was found significant. The effect of the difference between treatments on t50 values was also examined, and the ranking from most effective treatment was temperature pretreatment, control and ethanol pretreatment. Accordingly, although the effect of pretreatment was determined as significant for t50 values, it was insignificant for the t90 values. As regards fructose consumption, t50 and t90 were found in the range 30.84–36.83 and 45.19–54.42 h, respectively, for control samples, 31.59–42.03 and 47.19–54.50 h for temperature-pretreated cells and 30.26–41.58 and 50.70–74.83 h for ethanol-pretreated cells.

The effect of the type of yeast strain on t50 and t90 values was found significant (*p* < 0.05), while the type of pretreatment was significant only for t50. Moreover, the interaction effect of type of strain x type of pretreatment was not significant for t50 and t90 values (*p* ˃ 0.05). The highest t50 value was related to strain 2 and was found different from all the other strains (*p* < 0.05); moreover, the difference among the fructose t50 values of the other strains was determined to be insignificant (*p* > 0.05). The highest t90 values were found to belong to strain 2, significantly different from all other strains (*p* < 0.05). Ordering the strains from the most performing (fructose consumption) to the least efficient, the rank for t90 is as follows: S122, 47, 7 and 2. The ranking from most effective treatment was similar for glucose, i.e., temperature pretreatment, control and ethanol pretreatment. 

From an overall look at the results, a mild temperature pretreatment of yeast cells allowed for an increase in glucose consumption rate by strains 2, 7 and 47 (t90 reduced by around 8–10% with respect to control); ethanol pretreatment produced a similar effect, with the exception of strain 2. Interestingly, temperature pretreatment allowed for an increase in fructose consumption rate by the strains 7 and 47, evidenced by a t90 value 10% lower than the control; ethanol pretreatment reduced the t90 values by only 5%. 

## 4. Discussion

Several studies have highlighted that yeasts possess a slightly higher preference for glucose than fructose consumption during wine fermentations; so far, residual sugars in fermented grape must usually contain more fructose than glucose [7,8]. Nevertheless, a high residual fructose/glucose ratio may contribute to sluggish and stuck fermentations, a major problem in the global wine industry [9]. In the present research, we have investigated whether pretreatment of four wine yeasts (three of them indigenous and one reference strain) before fermentation in synthetic must can affect glucose and fructose consumption, with particular focus on the last sugar. 

Throughout the fermentation period, yeasts are exposed to several stresses. Extreme conditions may lead to a reduction in growth speed and survival rate, and therefore, cells tend to reduce fermentation efficiency, depending on the severity of the vinification procedures [19]. Nevertheless, some studies suggested that when cells are exposed to mild stresses, they undergo cellular innovation and maintain close-to-optimal conditions within the organism as a whole [1,2,3,23]; in this condition, the faster yeast strain able to adapt to changes in the environment will probably become the dominant strain during the winemaking process [24]. 

In the present study, results showed that overall only one strain showed an increase in glucose uptake, and two of the four strains showed an increase in fructose uptake, highlighting that they are sensitive and tend to respond quickly to the mild stress applied than the other ones. This situation can be explained by a more effective activation of the cellular machinery to control stress condition, involving the rapid synthesis of protective molecules and the activation of signal transduction systems, which induce secondary events such as the activation of pre-existing enzyme activities and the transcription of genes encoding factors having protective functions [25]. Additionally, a correlation exists between strain resistance, fermentative behavior and, under some conditions, the expression of some stress-induced genes. Guillaume et al. [26] investigated the molecular basis of this enhanced fructose utilization capacity by studying the properties of several hexose transporter (HXT) genes and found that the higher fructose utilization capacity of selected wine yeast strain results from the expression of an allelic variant of HXT3.

The wine yeast strains used in the present research consumed glucose more than fructose. Similar findings were obtained by Berthels et al. [7]: they studied the discrepancy between glucose and fructose utilization by sixteen *S. cerevisiae* and one *S. bayanus* during wine fermentations. Their research showed that all tested strains consumed glucose more rapidly than fructose, confirming the characteristics of *Saccharomyces* wine yeast strains. Similarly to our study, even though the fermentation started with approximately equal amounts of the two sugars, the concomitant but slower fructose utilization led to a discrepancy between the glucose and fructose levels during the early phase of the fermentation.

The reasons for the differences in the rates of glucose and fructose utilization seem to be correlated with the first steps in hexose metabolism, particularly prior to the formation of fructose-1,6-bisphosphate [1], in particular sensing extracellular sugars, their transport across the plasma membrane and phosphorylation [27]. Berthels et al. [9] showed that discrepancies in glucose/fructose consumption were related to different hexokinase kinetic properties of *Saccharomyces* strains. Likewise, Viana et al. [28] highlighted that *S. cerevisiae* wine yeast strains consume fructose in longer period than glucose in a synthetic medium. Díaz-Hellín et al. [1] declared that glucose and fructose are transported by the same hexose transporters (HXT), which present a greater affinity for glucose, so that fructose becomes the predominant residual sugar during fermentation. They investigated the relation between HXT3 gene expression and fructose/glucose discrepancy with a commercial wine strain of *S. cerevisiae* and reported that a correlation between fructose/glucose discrepancy and HXT3 gene expression is present. However, Guillaume et al. [26] found a *S. cerevisiae* wine yeast strain with a high fructose utilization ability due to a mutated HXT3 allele, meaning that fructose consumption could be altered by the expression of a mutated hexose transporter. 

The statistical analysis carried out on the here reported data evidenced that the increase or change in the amount of consumed glucose or fructose did not cause the same increase or change in the amount of ethanol produced. This behavior needs to be investigated separately for each strain. Mannazzu et al. [29] reported that the relationship between the amount of sugar consumption and ethanol production of three *S. cerevisiae* strains in a synthetic medium is not linear, i.e., the strains’ ranking for sugar consumption did not reflect the same ranking in terms of ethanol production.

The obtained results also evidenced that the effect of the interaction between the type of treatment and the type of strain on glucose and fructose consumption was statistically significant: in particular, glucose consumption by ethanol-pretreated 2, 47 and S122 strains significantly increased compared to control samples (*p* < 0.05). In temperature pretreatments, strain 7 consumed the highest level of glucose, while strain S122 consumed the lowest; applying ethanol pretreatments, strain 2 was the top-consuming, while strain 7 was the lowest glucose-consuming yeast.

As regards fructose consumption, strains 2 and S122 decreased their performance when temperature pretreatment was applied, while interestingly, strain 7, and also 47 even if with lower efficiency, improved both fructose consumption (g/L) and its consumption rate, evidenced with a low t90 value. *S. cerevisiae* strain 7 can, thus, be considered a promising strain to be subjected to a mild temperature treatment (45 °C, 1 h) prior to the fermentation step to reduce residual fructose in wine.

Literature data reported on the use of other pretreatments for wine yeasts: Mattar et al. [30] examined the effect of pulsed electrical fields (PEF) as pretreatment for glucose and fructose consumption by *S. cerevisiae* wine strains in synthetic fermentation media at 30 °C. They reported that electro-stimulation improved fermentation characteristics by increasing yeast metabolism; in particular, fructose consumption in samples with electrically activated inoculum exceeded two times that of the control samples. Nevertheless, this kind of pretreatment cannot be easily applied in fermentation plants.

Modelling a fermentation through a suitable mathematical model will be helpful for process design and control purposes [31]. Recent studies show that Gompertz or sigmoidal models (developed by the Gompertz model) can efficiently describe the fermentation process. Tronchoni et al. [19] studied 12 strains belonging to the species *S. cerevisiae*, *S. bayanus var. uvarum* and *S. kudriavzevii*: fermentations were carried out at 12 and 28 °C comparatively, and the suitability of three different mathematical models for process description was investigated. Results showed that glucose and fructose consumption at 12 °C was generally well defined by the sigmoidal model. 

Other authors declared that the Gompertz equation described sugar consumption and alcohol production during microfermentation trials very well, with R^2^ values higher than 0.98 [2,32]. In contrast, O’Neill et al. [33] reported that the Gompertz model was weak in modelling substrates consumption, especially in the late parts of fermentation. Our obtained goodness fit parameters showed the sigmoidal model to be the most suitable model, as it is able to describe glucose and fructose consumption in a fermentation environment with higher R^2^, lower RMSE, MBE and χ^2^ values with respect to the linear and exponential decay functions.

## 5. Conclusions

In this study, the effect of two different pretreatments on glucose and fructose consumption applied to four different *S. cerevisiae* wine strains prior to fermentation was examined. Results highlighted that the amount of consumed substrate or produced ethanol depends either on the type of pretreatment and the used strain. Among the tested wine yeast, *S. cerevisiae* Narince 3 (coded 7) can be selected for the prosecution of the research, due to its interesting characteristic of increasing fructose consumption after a mild temperature pretreatment. Although, in general, the commercial wine yeasts are known to be more tolerant to stress conditions than the indigenous yeasts, this study showed that a temperature pretreatment may be effective on indigenous yeasts in preventing undesirable sweetness in wines and arrested or so-called stuck fermentation by increasing sugar consumption. Indeed, it must be taken into consideration that in transferring the obtained results into the industry, the complexity of must composition as well as its variation in composition across batches can indeed impact the success of the application of the fermentation models here reported.

Future work will be aimed at understanding the ability of this strain to withstand these yeasts under real fermentation conditions to validate the preliminary results, paying special attention to their fermentative properties in grape must. Indeed, several authors have highlighted that a mild heat stress in yeasts not only increases resistance and survival to more severe heat exposure, but also increases protection against osmotic or oxidative stress [5,34]. 

These results will pave the way to the use of indigenous yeasts with improved characteristics to produce wines with a lower residual fructose level, also opening up new possibilities for the use of pretreated strains with higher fructose utilization capacities in other fermentation experiments.

## Figures and Tables

**Figure 1 foods-10-01129-f001:**
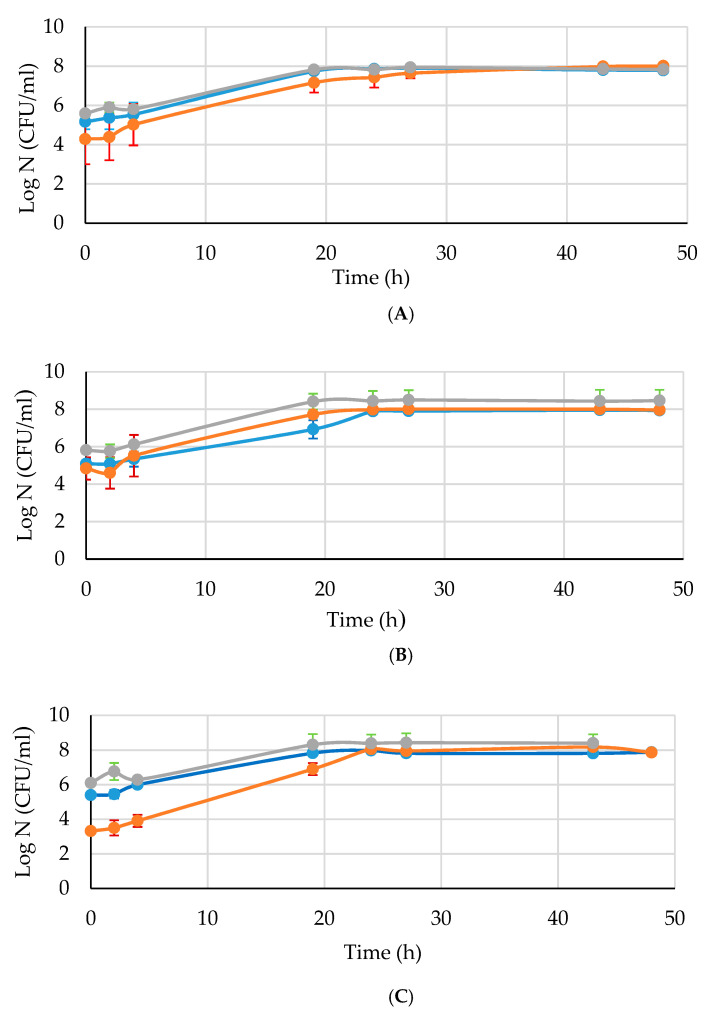
Growth curve of the strains 2, 7, 47 and S122 in synthetic medium. Symbols indicate different samples as blue circle (control), orange circle (temperature-pretreated) and grey circle (ethanol-pretreated). (**A**) *S. cerevisiae* 2; (**B**) *S. cerevisiae* 7; (**C**) *S. cerevisiae* 47; (**D**) *S. cerevisiae* S122.

**Figure 2 foods-10-01129-f002:**
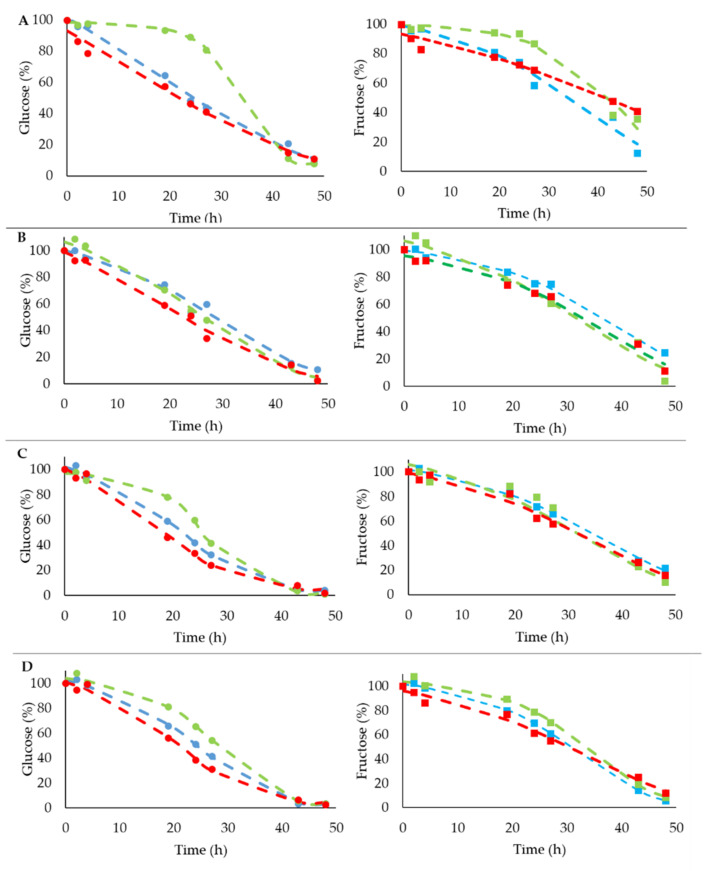
Observed and predicted with sigmoidal model (dashed lines) data related to glucose (left, circles symbols) and fructose (right, square symbols) by four strains after different pretreatments. (**A**) Strain coded 2, (**B**) strain coded 7, (**C**) strain coded 47, (**D**) strain coded S122. Type of pretreatments showed with different colors: blue—control, green—temperature, red—ethanol pretreatments.

**Table 1 foods-10-01129-t001:** Glucose and fructose consumption as well as ethanol concentrations (g/L) at different intervals during fermentation performed by *S. cerevisiae* strains coded 2, 7, 47 and S122 without (control) or with pretreatments (temperature or ethanol). Data presented as mean ± standard deviation, two replicates were analyzed.

Yeast Strain	Pretreatment	Time Distance (h)	Glucose (g/L)	Fructose (g/L)	Ethanol (g/L)
2	None (control)	0–19	39.01 ± 6.36	18.79 ± 7.09	21.65 ± 1.45
		0–24	57.23 ± 4.34	28.48 ± 7.46	30.65 ± 8.49
		0–48	95.08 ± 5.93	75.38 ± 13.40	66.93 ± 9.93
	Temperature	0–19	10.68 ± 4.42	2.30 ± 2.30	7.68 ± 7.08
		0–24	33.33 ± 29.45	14.53 ± 12.27	18.63 ± 18.63
		0–48	89.50 ± 16.26	60.18 ± 8.10	58.70 ± 0.01
	Ethanol	0–19	60.50 ± 16.05	27.53 ± 1.52	24.05 ± 10.18
		0–24	76.63 ± 14.04	34.10 ± 1.56	34.72 ± 2.14
		0–48	127.35 ± 7.00	73.93 ± 1.38	76.50 ± 1.27
7	None (control)	0–19	22.28 ± 2.30	14.15 ± 1.98	5.33 ± 5.33
		0–24	41.00 ± 15.63	20.40 ± 2.19	20.20 ± 15.13
		0–48	99.75 ± 0.95	83.15 ± 15.27	92.43 ± 23.43
	Temperature	0–19	31.23 ± 1.03	24.35 ± 5.73	9.75 ± 2.76
		0–24	47.35 ± 2.62	32.40 ± 6.65	26.63 ± 8.03
		0-48	103.18 ± 7.81	98.15 ± 2.83	71.78 ± 9.93
	Ethanol	0–19	32.60 ± 1.34	19.53 ± 0.11	18.05 ± 3.82
		0–24	39.40 ± 11.03	24.53 ± 9.58	26.93 ± 5.34
		0–48	77.53 ± 9.86	66.73 ± 6.33	70.45 ± 19.45
47	None (control)	0–19	43.70 ± 2.48	17.08 ± 1.10	11.65 ± 11.65
		0–24	38.04 ± 18.23	29.98 ± 6.19	34.95 ± 5.02
		0–48	102.18 ± 6.98	83.10 ± 18.74	76.55 ± 3.96
	Temperature	0–19	20.85 ± 16.12	11.14 ± 9.03	11.67 ± 6.76
		0–24	38.04 ± 12.89	19.56 ± 11.83	22.35 ± 12.23
		0–48	91.36 ± 5.96	84.24 ± 2.05	75.89 ± 3.80
	Ethanol	0-19	60.93 ± 1.31	18.58 ± 13.47	21.53 ± 21.53
		0–24	75.45 ± 2.26	39.33 ± 1.66	47.48 ± 1.52
		0–48	110.88 ± 5.27	87.68 ± 6.97	87.30 ± 3.68
S122	None (control)	0–19	32.13 ± 0.74	19.30 ± 0.85	20.80 ± 0.42
		0–24	45.58 ± 0.32	28.98 ± 0.74	33.38 ± 1.80
		0–48	90.05 ± 1.95	90.45 ± 1.98	80.25 ± 4.45
	Temperature	0–19	4.92 ± 2.58	1.15 ± 0.85	6.15 ± 2.76
		0–24	22.48 ± 8.38	14.08 ± 6.12	13.85 ± 5.80
		0–48	80.58 ± 6.89	77.73 ± 6.26	71.33 ± 1.31
	Ethanol	0–19	48.35 ± 16.48	23.50 ± 6.51	34.23 ± 8.87
		0–24	67.33 ± 13.75	38.60 ± 2.69	40.85 ± 7.50
		0–48	105.85 ± 12.37	87.93 ± 6.61	85.30 ± 0.85

**Table 2 foods-10-01129-t002:** Effect of different variables (type of pretreatment—TP, type of strain—TS and their interaction) on glucose and fructose consumption and ethanol production.

Source	Dependent Variable	df	Sum of Square	F
Type of Pretreatment (TP)	Glucose ^a^	2	3986.344	36.684 ***
	Fructose ^b^	2	453.384	7.247 **
	Ethanol ^c^	2	1257.563	14.136 ***
Type of Strain (TS)	Glucose	3	776.018	7.141 **
	Fructose	3	142.591	2.279
	Ethanol	3	165.783	1.863
TP × TS	Glucose	6	933.023	8.586 ***
	Fructose	6	439.423	7.024 ***
	Ethanol	6	130.316	1.465

^a^ R^2^ = 0.947 (adjusted R^2^ = 0.986); ^b^ R^2^ = 0.965 (adjusted R^2^ = 0.931); ^c^ R^2^ = 0.944 (adjusted R^2^ = 0.889); *** *p* ˂ 0.001; ** *p* ˂ 0.05.

**Table 3 foods-10-01129-t003:** Effect of interaction between type of pretreatment and type of strain on glucose, fructose levels and effect of treatment difference on ethanol level.

Pretreatment	Yeast Strain	Glucose (g/L)	Fructose (g/L)	Ethanol (g/L)
None (control)	2	63.77 ^abcde*^	40.88 ^ab^	41.23 ^a^
	7	54.34 ^cdef^	39.23 ^ab^	
	47	69.38 ^abcd^	43.84 ^ab^	
	S122	55.92 ^cdef^	46.24 ^ab^	
Temperature	2	44.50 ^ef^	25.67 ^b^	32.87 ^a^
	7	60.58 ^cde^	51.63 ^a^	
	47	50.08 ^def^	38.31 ^ab^	
	S122	35.99 ^f^	30.98 ^b^	
Ethanol	2	88.159 ^a^	45.18 ^ab^	47.28 ^b^
	7	49.84 ^def^	36.93 ^ab^	
	47	82.42 ^ab^	48.53 ^a^	
	S122	73.84 ^abc^	50.01 ^a^	

* Letters represent similarities or differences within each dependent variable. In the Tukey test, the effect of interaction between type of strain × type of pretreatment on the changes in glucose and fructose levels, and the effect of only the type of pretreatment was significant on ethanol production, only the significant values were given in the table.

**Table 4 foods-10-01129-t004:** Model and goodness of fit parameters (GOF) of the different models for glucose and fructose consumption (* if the value is lower than 10^−4^, it was given as zero (0). ** ND: not detected, model not suitable).

Strain	Treatment		Linear				Exponential Decay Function				Sigmoid Function			
			R^2^	RMSE	MBE	χ^2^	R^2^	RMSE	MBE	χ^2^	R^2^	RMSE	MBE	χ^2^
2	Control	Glucose	0.991	3.225	0.002	20.80	0.994	2.698	0 *	14.56	0.994	2.505	0.0 *	12.55
	Temperature		0.806	16.144	0.004	521.23	0.763	21.441	11.910	919.43	0.999	0.888	0 *	1.58
	Ethanol		0.981	4.169	0 *	34.76	0.987	3.599	0.854	0.987	0.986	3.631	0 *	26.36
7	Control		0.961	6.192	0.003	76.69	0.961	4 × 10^5^	4 × 10^5^	3 × 10^9^	0.989	3.594	0 *	25.83
	Temperature		0.989	3.913	0.003	30.62	0.989	3.922	0 *	30.77	0.993	3.303	0 *	21.82
	Ethanol		0.988	3.864	0 *	29.85	0.988	13.023	0 *	339.19	0.991	3.290	0 *	21.65
47	Control		0.979	5.596	0 *	62.64	0.990	3.740	0 *	27.98	0.998	1.664	0 *	5.53
	Temperature		0.965	7.001	0 *	98.02	0.965	7.008	0 *	98.22	0.997	2.141	0 *	9.17
	Ethanol		0.948	8.701	0 *	151.40	0.990	3.824	0 *	29.25	0.996	4.524	1.712	40.94
S122	Control		0.990	3.828	0.003	29.30	0.990	3.807	0 *	28.99	0.999	1.390	0 *	3.86
	Temperature		0.963	7.437	0.008	110.63	0.963	7.445	0 *	110.86	0.996	2.313	0 *	10.70
	Ethanol		0.974	6.148	0 *	75.59	0.989	3.976	0 *	31.62	0.997	2.252	0 *	10.14
2	Control	Fructose	0.946	6.836	0 *	93.47	0.946	8.140	4.411	132.52	0.977	4.462	0 *	39.82
	Temperature		0.795	11.521	0 *	265.47	0.795	14.021	7.988	393.18	0.972	4.297	0.101	36.93
	Ethanol		0.955	4.000	0.006	32.00	0.955	6.220	0 *	77.38	0.964	3.568	0 *	25.46
7	Control		0.946	12.152	0 *	295.36	ND **	6 × 10^4^	0 *	7 × 10^7^	0.994	2.207	0.020	9.74
	Temperature		0.795	20.182	11.94	814.60	0.794	19.683	11.075	774.86	0.973	5.709	0 *	65.19
	Ethanol		0.955	12.513	6.230	313.16	0.930	16.115	4.640	519.40	0.988	3.260	0 *	21.25
47	Control		0.966	5.504	0 *	60.60	0.965	5.511	0 *	60.74	0.996	1.990	0 *	7.92
	Temperature		0.893	10.633	0.005	226.10	0.893	10.636	0 *	226.27	0.995	2.266	0 *	10.27
	Ethanol		0.970	5.253	0.008	55.18	ND **	ND **	16.484	937.79	0.988	3.353	0.032	22.48
S122	Control		0.960	7.126	0 *	101.56	0.959	7.133	0 *	101.77	0.999	1.007	0 *	2.03
	Temperature		0.916	10.262	0.007	210.63	0.916	10.267	0 *	210.82	0.996	2.323	0 *	10.79
	Ethanol		0.976	4.700	0 *	44.17	0.975	4.707	0 *	44.31	0.988	3.295	0 *	21.71

**Table 5 foods-10-01129-t005:** Equations (sigmoidal model) to estimate the residual glucose and fructose content (g/L) in synthetic must fermented by the tested yeast strains either untreated (control) and subjected to temperature or ethanol pretreatment. The time needed to consume 50% (t50) and 90% (t90) of the initial sugar concentration in each condition is also given.

Yeast Code	Treatment	Sugar	Model Equations	t_50_ (h)	t_90_ (h)
2	Control	Glucose	$Y=1.017+169.069\times e^{-e^{0.035\times \left(t-18.493\right)}}$	24.60	49.22
	Temperature		$Y=7.764+90.767\times e^{-e^{0.173\times \left(t-36.144\right)}}$	34.60	43.71
	Ethanol		$Y=185.558\times e^{-e^{0.029\times \left(t-12.601\right)}}$	21.84	49.13
7	Control		$Y=4.354+111.649\times e^{-e^{0.062\times \left(t-30.259\right)}}$	28.46	47.92
	Temperature		$Y=131.263\times e^{-e^{0.058\times \left(t-27.006\right)}}$	26.40	43.32
	Ethanol		$Y=148.553\times e^{-e^{0.043\times \left(t-20.633\right)}}$	22.60	43.51
47	Control		$Y=4.224+114.825\times e^{-e^{0.080\times \left(t-22.717\right)}}$	21.67	36.39
	Temperature		$Y=2.518+95.311\times e^{-e^{0.167\times \left(t-27.801\right)}}$	25.64	33.39
	Ethanol		$Y=4.794+127.747\times e^{-e^{0.070\times \left(t-17.836\right)}}$	18.38	34.52
S122	Control		$Y=2.542+112.011\times e^{-e^{0.083\times \left(t-26.111\right)}}$	24.27	38.16
	Temperature		$Y=3.802+103.392\times e^{-e^{0.111\times \left(t-30.005\right)}}$	28.06	39.33
	Ethanol		$Y=4.341+113.642\times e^{-e^{0.081\times \left(t-22.060\right)}}$	20.93	35.57
2	Control	Fructose	$Y=110.097\times e^{-e^{0.059\times \left(t-38.140\right)}}$	34.11	53.05
	Temperature		$Y=0.056+1101.118\times e^{-e^{0.095\times \left(t-45.688\right)}}$	42.03	54.50
	Ethanol		$Y=116.558\times e^{-e^{0.032\times \left(t-46.788\right)}}$	41.58	74.83
7	Control		$Y=0.004+106.313\times e^{-e^{0.065\times \left(t-41.166\right)}}$	36.83	54.42
	Temperature		$Y=124.357\times e^{-e^{0.0555\times \left(t-33.267\right)}}$	31.59	49.92
	Ethanol		$Y=104.864\times e^{-e^{0.062\times \left(t-37.830\right)}}$	33.02	51.52
47	Control		$Y=0.338+113.638\times e^{-e^{0.057\times \left(t-37.915\right)}}$	34.61	53.70
	Temperature		$Y=4.554+96.017\times e^{-e^{0.099\times \left(t-37.752\right)}}$	34.81	48.44
	Ethanol		$Y=0.027+116.047\times e^{-e^{0.053\times \left(t-34.925\right)}}$	31.69	51.88
S122	Control		$Y=1.105+107.332\times e^{-e^{0.080\times \left(t-33.836\right)}}$	30.84	45.19
	Temperature		$Y=4.767+102.063\times e^{-e^{0.096\times \left(t-35.821\right)}}$	33.67	47.19
	Ethanol		$Y=113.085\times e^{-e^{0.053\times \left(t-34.075\right)}}$	30.26	50.70

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
