# Peer review of "Mild Pretreatments to Increase Fructose Consumption in *Saccharomyces cerevisiae* Wine Yeast Strains"

_foods, 2021, doi:10.3390/foods10051129_

Round 1

Reviewer 1 Report

The article investigates the benefit and modeling of sugar consumption or ethanol production due to thermal or ethanolic stress of Saccharomyces cerevisiae in wine quality.

The manuscript is original and has high quality. The results, discussion and conclusions are sufficiently argued and justified with the data. The data and conclusions are important for the wine sector, and the area of food science and technology. The materials and methods allow the results to be reproduced. The experiments fit a suitable design well. References are correct in the text and in the reference list. The manuscript is easy and fast to read. No relevant plagiarism is detected, only on lines 163 and 440-442 (https://plagiarismdetector.net/es). The English language is understandable and correct.

However, to contribute to the improvement of the article, I propose some changes that are detailed below.

- In lines 270-273, the effect of the treatments on the t50 value for glucose is analyzed, but the t90 values have not been commented… And in lines 281-284, the effect of the treatments on the t90 value for fructose is analyzed, but the t50 values have not been commented. It is suggested that the comments of these data are complemented.

- Mistakes in the data: where 95.05 is shown, 90.05 must be written (line 193); and where 0.991 is observed, 0.986 must be written (line 224).

- In line 182-183, where it says “to reach the stationary phase at 19 h”, it should be written before 19 h” or “between 4-19 h” since there is no more data from around the fourth hour and it is not modeled.

- In Table 1, it is recommended to add that the data are presented as the mean ± standard deviation and the number of replicates analyzed. Also, it can see some data in which the standard deviation is equal to the mean... is this correct?

- In Table 4, the data for strain 2> control> sigmoid function> MBE appears to be a mistake. On the other hand, the ethanol production data is not presented.

- It is recommended to italicize the terms of the equations to facilitate their reading: lines 148-150, 152-154, 157-159 and 168-169. In line 153, the term ... is it Kexp or K?

- It is recommended that the figures and tables were approximated in a position closer to their explanation in the manuscript. And merge paragraph 274-276 with paragraph 277-284, since they are the same idea.

- It is suggested coherence between American or British English: modeling/modelling (line 75, 142, 226, 395, and 406), analyses/analyzes (line 137-138, and 434), color (line 300), behavior/behaviour (line 370, and 338).

- It is suggested coherence between unit abbreviations: figure 1 shows Log N (cfu/ml), while the rest of the manuscript shows Log CFU/mL (line 134). In all the text there is a space between the data and its units, except in lines 140, 399 and 401. In figure 2, “hour” is observed, while in the rest of the manuscript “h”.

- It is suggested coherence between the probability p format: lines 206, 218, 219, 240, 267, 277, 280, 281, and 378. Also, in line 456 put the + charge of the hydrogen as a superscript.

- In line 174, is it Statistica or Statistical? In line 233, the "and" is missing in the list.

Reviewer 2 Report

The manuscript “Mild pre-treatments to increase fructose consumption in Saccharomyces cerevisiae wine yeast strains” need minor correction:

  • In reference 19 "Technological properties of indigenous wine yeast strains isolated from wine production regions of Turkey" it is stated that the A1 strain is Saccharomyces bayanus, while in the manuscript has been written that it is S. cerevisiae. Please, correct it.
  • Making 2 replicates of the analysis is not sufficient and the Tukey test is unreliable
  • Please mark the statistically significant differences in tab. 1.

Reviewer 3 Report

Overall, this is an interesting paper and there are some good results. I would like to see further information on typical wine must composition within the introduction, particularly nitrogen due to the importance of YAN concentration (a table would be useful). This would help to explain the composition of the synthetic medium more clearly – including why 115g/L of each sugar was chosen. Some further discussion of the impact of total sugar concentration (increasing initial gravity) on the uptake of individual sugars would also be good. Further discussion on the complexity of must composition, variation in composition across batch-to-batch, and variation in fermentation conditions, and the impact these have in applying fermentation models would be useful too.

There is no mention of the number of replicates. This should be included in the methods and throughout the results section.

Further comments/corrections:

Abstract

Saccharomyces in full on first use – Line 11.

Remove coding references – Line 13, ‘(1-2-7)’ and ‘(S122)’.

Acronyms should be given in full on first use – Line 17, R2/RMSE/MBE.

Replace strain references (7&47) from line 19 with “by two strains”.

Replace “S. cerevisiae strain 7” with “ One of the indigenous strains showed particular promise for mild temperature treatment….”.

Introduction

Line 30: ‘Proper’ needs definition or re-wording – ‘efficient’?

Line 45-46: ‘With fructose being the predominant residual sugar’.

Line 46: Insert ‘The’ before ‘dominant’.

Line 83: Insert ‘the’ before ‘fermentation’.

Methods

Why preheat at 60°C before moving to 45°C? Would this be practical in an industrial set up?

For ethanol pre-treatment, why 14%, and what was the stock concentration of ethanol added? Was it grain neutral spirit? Why immediate transfer to the must rather than a set time period?

Further detail needed in the methods, for example a reference to further information on HPLC.

What was the yeast pitching rate? Was it typical?

Results

Line 166-167: Equation 5 – ‘MBE’.

Line 181: Remove ‘s’ from ‘yeasts’.

Line 217: Insert ‘to be’ within ‘found inversely’.

Line 270: Insert ‘to be’ within ‘found significant’.

Line 289: Insert ‘s’ in ‘strains’.

Line 322: Replace ‘all’ with ‘the’.

Discussion

Line 330: The statement ‘temperature pre-treatment caused an increase in glucose and fructose consumption’ is incorrect as the results show that overall only one strain showed an increase in glucose uptake and two of the four strains showed an increase in fructose uptake.
